# Integrating the Transcriptome and Proteome to Postulate That TpiA and Pyk Are Key Enzymes Regulating the Growth of *Mycoplasma Bovis*

**DOI:** 10.3390/microorganisms12102012

**Published:** 2024-10-03

**Authors:** Fei Yang, Mengmeng Yang, Fan Liu, Yanrong Qi, Yanan Guo, Shenghu He

**Affiliations:** 1Institute of Animal Sciences, Ningxia Academy of Agricultural and Forestry Sciences, Yinchuan 750002, China; yangfeiweiwuxian@126.com (F.Y.); 15809605110@163.com (M.Y.); 2College of Animal Science and Technology, Ningxia University, Yinchuan 750021, China; liuf2785@163.com (F.L.); hlxxmcyz@126.com (Y.Q.)

**Keywords:** *Mycoplasma bovis*, prokaryotic transcriptome, 4D-label-free quantitative proteomics, TpiA, Pyk

## Abstract

*Mycoplasma bovis* is a global problem for the cattle industry due to its high infection rates and associated morbidity, although its pathophysiology is poorly understood. In this study, the *M. bovis* transcriptome and proteome were analyzed to further investigate the biology of clinical isolates of *M. bovis*. A differential analysis of *M. bovis*, a clinical isolate (NX114), and an international type strain (PG45) at the logarithmic stage of growth, was carried out using prokaryotic transcriptome and 4D-label-free quantitative non-labeled proteomics. Transcriptomics and proteomics identified 193 DEGs and 158 DEPs, respectively, with significant differences in 49 proteins/34 transcriptomic CDS post-translational protein sequences (15 jointly up-regulated and 21 jointly down-regulated). GO comments indicate membrane, cytoplasmic and ribosome proteins were important components of the total proteins of *M. bovis* NX114 clinical isolate. KEGG enrichment revealed that *M. bovis* NX114 is mainly associated with energy metabolism, the biosynthesis of secondary metabolites, and the ABC transporters system. In addition, we annotated a novel adhesion protein that may be closely related to *M. bovis* infection. Triosephosphate isomerase (TpiA) and Pyruvate kinase (Pyk) genes may be the key enzymes that regulate the growth and maintenance of *M. bovis* and are involved in the pathogenic process as virulence factors. The results of the study revealed the biology of different isolates of *M. bovis* and may provide research ideas for the pathogenic mechanism of *M. bovis*.

## 1. Introduction

*Mycoplasma bovis* is a pathogenic microorganism that causes a wide range of diseases in cattle, including keratoconjunctivitis, mastitis, endometritis, otitis media, arthritis, pneumonia, and respiratory syndromes, and it has caused serious economic losses to the global cattle industry [1,2,3]. The lack of a cell wall on the surface of mycoplasmas and its resistance to beta-lactams and other antimicrobial compounds limit the effectiveness of antibiotic therapy [4]. The emergence of antibiotic resistance has also hampered the treatment of the disease, leading to the emergence of more resistant *M. bovis* isolates, especially to fluoroquinolones, doxycycline, and enrofloxacin antibiotics [5]. There is also a lack of effective vaccines to prevent *M. bovis* infections, so prevention and control of the disease are currently extremely difficult.

Proteins are important carriers of cellular functions and life activities and are the final form of expression of genetic information within microorganisms. The surface of mycoplasmas lacks cell wall structures, and *M. bovis* lacks tip structures for adhesion to host cells, like those specialized for *Mycoplasma pneumoniae* and *Mycoplasma genitalium* [6]. Therefore, it is known that *M. bovis* membrane proteins and membrane-associated proteins are crucial for interaction with host cells. It can play a role in the process of infecting the host by participating in bacterial adhesion, virulence-associated factors, mediators of cell signaling, effectors that modulate host immunity, and immunogens [7,8]. Thus, *M. bovis* proteins play a role in inducing host pathogenicity and immune response throughout the infection process. In addition, numerous studies have analyzed the secreted proteins of various mycoplasmas by untagged proteomics, isotope-tagged relative and absolute quantification (iTRAQ^TM^), and liquid chromatography–tandem mass spectrometry (LC-MS/MS) [9,10,11,12]. Meanwhile, transcriptomics was used to identify important phosphatases involved in the glycolytic pathway in *M. bovis* [13]. Proteomics was used to comparatively analyze differentially secreted proteins of strongly virulent and attenuated strains of *M. bovis*, and advances have been made in pathogenic mechanisms [14].

Transcriptomes and proteomes are closely related to both upstream and downstream genomics. Merging transcriptome and proteome analyses will help to obtain a panoramic view of *M. bovis* in terms of gene expression and regulation. In this study, we analyzed the homologous and unique proteins between *M. bovis* clinical isolate NX114 (pneumonia isolate) and international standard strain PG45 (mastitis isolate) using a combination of prokaryotic transcriptomics and 4D-label-free quantitative proteomics. The goal is to further investigate the *M. bovis* proteome’s virulence, resistance, or immunological properties in order to elucidate its pathogenesis or identify effective drug or vaccine candidates.

## 2. Materials and Methods

### 2.1. Bacterial Strains and Media

*M. bovis* clinical strain NX114 (GenBank accession no. CP135997) was obtained from diseased Holstein cows in China, while *M. bovis* international type strain PG45 was generously contributed by Professor Aizhen Guo of Huazhong Agricultural University in China. The two strains were preserved in PPLO broth with 40% glycerol at −80 °C in a laboratory before being cultured in PPLO [15] broth before use.

### 2.2. Transmission Electron Microscopy

Samples were first treated with a 3% glutaraldehyde solution, then postfixed with 1% osmium tetroxide. They were then dehydrated using a series of acetone solutions, infiltrated with Epox 812 for an extended period, and finally embedded. The semithin sections were stained with methylene blue, while the ultrathin sections were cut using a diamond knife and stained with uranyl acetate and lead citrate. The sections were analyzed using a JEM-1400-FLASH Transmission Electron Microscope.

### 2.3. Transcriptomics Analysis of M. Bovis

The Trizol Reagent (Invitrogen Life Technologies, Thermo Scientific, Bremen, Germany) was used to separate the total RNA. The determination of quality and integrity was conducted using a NanoDrop spectrophotometer (Thermo Scientific, Bremen, Germany) and a Bioanalyzer 2100 system (Agilent, Palp Alto, CA, USA). The Ribo-Zero rRNA Removal Kit (Illumina, San Diego, CA, USA) is used for mRNA sequencing. The first cDNA strand was synthesized using random oligonucleotides and SuperScript III. Subsequently, the synthesis of the second-strand cDNA was carried out utilizing DNA Polymerase I and RNase H. The remaining overhangs were transformed into blunt ends using exonuclease/polymerase activities, and the enzymes were then eliminated. Following the adenylation of the 3′ ends of the DNA fragments, the Illumina PE adapter oligonucleotides were ligated in order to facilitate hybridization. In order to choose cDNA fragments that are precisely 300 base pairs long, the library fragments were purified using the AMPure XP technology (Beckman Coulter, Beverly, CA, USA). DNA fragments including ligated adaptor molecules on both ends were specifically amplified using the Illumina PCR Primer Cocktail in a 15-cycle PCR reaction. The products underwent purification using the AMPure XP system and were quantified using the Agilent high sensitivity DNA test on a Bioanalyzer 2100 system (Agilent). The sequencing library was then analyzed using a NextSeq 500 platform (Illumina) by Shanghai Personal Biotechnology Cp., Ltd., Shanghai, China.

### 2.4. Label-Free Quantitative Proteomics Analysis of M. Bovis

Protein was extracted from *M. bovis* samples using SDT lysis buffer (4% SDS, 100 mMDTT, and 100 mM Tris-HCl at pH 8). The samples were cooked for 5 min, then ultrasonically sonicated and boiled for another 5 min. Cellular debris was removed by centrifugation at 16,000 g for 15 min. The supernatant was collected and measured using a BCA Protein Assay Kit (Bio-Rad, Hercules, CA, USA). The FASP technique, as reported by Wisniewski, Zougman et al. [16], was used to digest 200 μg of protein each sample. To prevent reduced cysteine, the detergent, DTT, and IAA were added to UA buffer. Finally, the protein solution was digested with trypsin (Promega) at a 50:1 ratio overnight at 37 °C. The peptide was obtained by centrifugation at 16,000× *g* for 15 min. The peptide was desalted with C18 StageTip for further LC-MS analysis. The Nanodrop One gadget measured peptide concentrations using OD280.

The LC-MS/MS analysis was conducted using a Q Exactive Plus mass spectrometer, which was connected to an Easy 1200 nLC system from Thermo Fisher Scientific. The peptide was first placed onto a trap column (100 μm × 20 mm, 5 μm, C18, Dr. Maisch GmbH, Ammerbuch, Germany) using buffer A (0.1% formic acid in water). The EASY-nLC system (Thermo Fisher Scientific, Bremen, Germany) was used to perform reverse-phase high-performance liquid chromatography (RP-HPLC) separation. A self-packed column (75 μm × 150 mm; 3 μm ReproSil-Pur C18 beads, 120 Å, Dr. Maisch GmbH, Ammerbuch, Germany) was employed at a flow rate of 300 nL/min. The mobile phase A used in RP-HPLC consisted of water containing 0.1% formic acid, whereas the mobile phase B consisted of 95% acetonitrile containing 0.1% formic acid. The peptides were separated over a period of 120 min using a gradual increase in the concentration of buffer B. The acquisition of MS data was performed using a data-dependent top20 approach, which dynamically selects the most abundant precursor ions from the survey scan (300–1800 m/z) for HCD fragmentation. The instrument was operated with the peptide recognition mode on. An internal standard with a lock mass of 445.120025 Da was used for mass calibration. The complete MS scans were obtained with a resolution of 70,000 at m/z 200 and 17,500 at m/z 200 for the MS scan. The maximum injection time for MS was set to 50 ms, and for MS/MS, it was also set to 50 ms. The collision energy was adjusted to a normalized value of 27, and the isolation window was set at 1.6 T. The length of the dynamic exclusion was 60 s.

The MS data were analyzed using MaxQuant 1.6.0.16 software. MS data were searched against the Uniprot-*Mycoplasmopsis bovis* (*Mycoplasma bovis*) (28903)-6273-20221207. Trypsin was chosen as a digestive enzyme. For database search, the maximum two missed cleavage sites and a mass tolerance of 4.5 ppm for precursor ions and 20 ppm for fragment ions were specified. The carbamidomethylation of cysteines was considered a fixed modification, while the acetylation of the protein’s N-terminal and methionine oxidation were characterized as variable modifications for database searching. The database search results were filtered and exported with <1% false discovery rate (FDR) at the pep-tide-spectrum-matched and protein levels, respectively. MaxQuant was used for label-free quantification, using the intensity determination and normalization algorithms published before. We computed the “LFQ intensity” of each protein in various samples as the best estimate, meeting all pairwise peptide comparisons, and found that this LFQ intensity was almost on the same scale as the summed peptide intensities. The quantitative protein ratios were weighted and adjusted using Maxquant’s median ratio. Proteins with a fold change ≥ 1.5-fold and a *p*-value < 0.05 were classified as substantially differentially expressed.

### 2.5. Bioinformatics Analysis

Bioinformatics data were analyzed using Perseus, Microsoft Excel, and R statistical computer tools. The pheatmap package, based on the open-source statistical language R25, was used to conduct hierarchical clustering analysis, using Euclidean distance as the distance metric and the complete approach as the agglomeration technique. To annotate the sequences, information was gathered from Uni-ProtKB/Swiss-Prot, the Kyoto Encyclopedia of Genes and Genomes (KEGG), and Gene Ontology. Fisher’s exact test was used for GO and KEGG enrichment analyses, along with FDR correction for multiple testing. GO terms were divided into three categories: biological processes (BPs), molecular functions (MFs), and cellular components (CCs). The enriched GO and Kegg pathways were statistically significant (*p* < 0.05). Protein–protein interaction (PPI) networks were also built using the STRING database and the Cytoscape software v.3.9.1.

### 2.6. Combined Proteomic and Transcriptomic Analysis of M. Bovis

The combined analysis of proteomics and transcriptomics includes statistics of common differential proteins or genes between proteomics and transcriptomics, a comparison between proteomics and transcriptomics, an expression cluster analysis (Cluster_Heatmap) and a differential multiple correlation analysis (FC_correlation). KEGG analysis (Total_KEGG) is also included. All analyses in this project were performed using R 4.2.2.

### 2.7. qRT-PCR Analyses

TRIzol reagent was used to isolate total RNA from the *M. bovis* samples. The RevertAid First Strand cDNA Synthesis Kit was used to make cDNA samples. qRT-PCR was performed using the FastStart Universal SYBR Green reagent and analyzed with the StepOnePlus RealTime PCR Systems (Applied Biosystems, Waltham, MA, USA). Gene expression was measured using the comparative threshold cycle technique (2^−ΔΔCt^) with GAPDH. The primer sequences utilized are shown in Appendix A.

## 3. Results

### 3.1. Transmission Electron Microscopy of M. Bovis

Transmission electron microscopy of the organelle structures of *M. bovis* NX114 and PG45 revealed a large number of ribosomes and soluble proteins in the cytoplasm of both strains (Figure 1).

### 3.2. Analysis of Prokaryotic Transcripts of M. Bovis

We compared the gene expression profiles of the *M. bovis* NX114 and PG45 groups using the Illumina platform to explore the gene profile of clinical isolate NX114. We then constructed and sequenced a total of six cDNA libraries, three from the NX114 group and three from the PG45 group. The total raw reads for the NX114 group were 15655472, 14150886, and 15689858, and the total raw reads for the PG45 group were 15509064, 15504262, and 15464574, respectively. After quality filtering (e.g., Q20, Q30, and the error rate), the clean reads were 14701712, 13306834, and 14770858 for the NX114 group, and 14770858, 14647256, and 14551518 for the PG45 group.

The RNA-seq of *M. bovis* NX114 and PG45 produced a total of 718 unigenes, and using BLAST software v.2.10.1 to annotate individual unigenes into 10 databases, a total of 718 unigenes (100%) were annotated in NR, 688 (95.82%) were annotated in eggNOG, 560 (77.99%) were annotated in eggNOG_Category, 453 (63.09%) were annotated in Swissprot, 419 (58.36%) were annotated in KEGG, 262 (36.49%) were annotated in Pathway, 530 (73.82%) were annotated by GO in Medium, 98 (13.65%) were annotated in by VFDB, 162 (22.56%) were annotated in by PHI, and 34 (4.74%) were annotated in by CARD (Figure 2A). FPKM density values showed gene expression profiles in *M. bovis* NX114 and PG45 (Figure 2B), and PCA analysis showed significant differences in the transcriptomes (Figure 2C). Based on the thresholds (|log2FoldChange| > 1 and *p*-value < 0.05), 193 DEGs were identified, of which 106 were up-regulated and 87 were down-regulated (Appendix A and Figure 2D). Specific databases showed that a total of 26 genes in the two *M. bovis* strains were simultaneously annotated in the CARH, VFDB, and PHI databases; 38 genes were found in both the VFDB and PHI, as well as 2 genes in both the CARH and PHI (Figure 3A). Among them, six unique genes (NX114:0, PG45:6) were annotated in the VFDB database and four unique genes (NX114:1, PG45:3) were annotated in the PHI database (Figure 3B). In addition, a total of 10 DEGs (3 up-regulated, 7 down-regulated) of *M. bovis* NX114 vs. PG45 were annotated in the CARH database, 26 DEGs (11 up-regulated, 15 down-regulated) were annotated in the VFDB database, and 42 DEGs (14 up-regulated, 28 down-regulated) were annotated in the PHI database, with a total of 5 DEGs (1 up-regulated, 4 down-regulated) being annotated in three databases at the same time (Figure 3C,D).

We performed GO analysis on 193 DEGs genes to better understand the underlying biological processes of *M. bovis*. We assigned the 193 DEGs to 135 GO categories (up-regulated DEGs accounted for 71 and down-regulated DEGs accounted for 64) (Appendix A). The GO enrichment analysis of DEGs showed that most of the highest enriched terms were MF and BP. The MF category mainly included catalytic activity (GO:0003824), carbohydrate derivative binding (GO:0097367), catalytic activity acting on RNA (GO:0140098) and ion binding (GO:0043167), and for the BP category, cellular component biogenesis (GO:0044085), small molecule biosynthetic process (GO:0044283) and phosphate-containing compound metabolic process (GO:0006796) were the top three enriched GO terms (Appendix A). The KEGG pathway annotation and enrichment analysis revealed the significant assignment of DEGs to 41 metabolic pathways (Figure 4A and Appendix A), with major enrichment in the glycolysis/gluconeogenesis (ko00010), purine metabolism (ko00230), beta-lactamase resistance (ko01501), citrate cycle (TCA cycle) (ko00020), nicotine and nicotinamide metabolism (ko00760), and RNA degradation (ko03018) pathways (Figure 4B). The protein–protein interaction analysis network was composed of 22 nodes and 120 edges, with nodes representing *M. bovis* proteins and edges representing connections. The proteins MBOVPG45_RS01305, MBOVPG45_RS01460, MBOVPG45_RS01385, and MBOVPG45_RS01465 are the core proteins of the network (Figure 4C).

### 3.3. Four-Dimensional Label-Free Quantitative Proteomics Analysis of M. Bovis

Four-dimensional label-free quantitative proteomics comparatively analyzed the protein characterization of *M. bovis*. The SDS-PAGE analysis revealed that the extracted *M. bovis* proteins had wide ranges of molecular weights. There were also notable differences in the protein bands between *M. bovis* strains NX114 and PG45, which allowed for further mass spectrometry analysis (Appendix A). Mass spectrometry analysis identified a total of 11,035 unique peptides and 908 proteins, with the peptide lengths primarily distributed in the range of 7 to 15 (Appendix A). Venn diagrams revealed 885 overlapping proteins among the 908 identified (Figure 5A). The PCA analysis showed that the NX114 and PG45 groups were separate, which means that there are important differences between the *M. bovis* groups (Figure 5B). Cluster analysis revealed similar protein abundance profiles within the same group of *M. bovis,* with significant differences observed between groups (Figure 5C). Differentially expressed proteins (DEPs) were screened for 58 up-regulated and 100 down-regulated proteins based on thresholds (FC > 1.5 or <1/1.5 and *p*-value < 0.05) (Appendix A and Appendix A).

The GO annotations of 158 DEPs revealed 93 functional terms at level 2, including 35 biological processes (BPs), 17 cellular components (CCs), and 42 molecular functions (MFs) (Figure 6A). The GO functional enrichment analysis of DEPs showed that the 20 most significantly enriched GOs were mainly MFs and BPs, mainly N-methyltransferase activity (GO:0008170), pseudouridine synthase activity (GO:0009982), DNA binding (GO:0003677), macromolecule modification (GO:0043412), RNA modification (GO:0009451) and pseudouridine synthesis (GO:0001522) (Appendix A and Appendix A). In addition, the KEGG pathway revealed 15 KEGG pathway components out of 158 DEPs (Appendix A), in which up-regulated DEPs were significantly enriched, mainly in metabolic pathways (mbv01100) and glycolysis/gluconeogenesis (mbv00010) (*p* < 0.05). Down-regulated DEPs were significantly enriched, mainly in homologous recombination (mbv03440) (Figure 6B).

The Protein Functional Interaction Network (PFIN) analysis was performed to construct significantly enriched pathways (*p* < 0.05) with significant differences in protein interactions. Metabolic pathways and glycolysis/gluconeogenesis (*p* < 0.05) were the core metabolic pathways (Figure 6C). The up-regulated DEPs AOA059Y438 (*p* = 2.7 × 10^−4^), AOA059XYN8 (*p* = 6.2 × 10^−5^), AOA059XZN7 (*p* = 9.2 × 10^−3^), AOA059XZl4 (*p* = 9.6 × 10^−4^), AOA059Y4A4 (*p* = 2.32 × 10^−5^), AOA059Y7N2 (*p* = 1.1 × 10^−3^) and AOA059Y900 (*p* = 2.3 × 10^−4^). The down-regulated DEPs AOA059Y8Q5 (*p* = 3.3 × 10^−4^) and AOA059Y8X4(*p* = 3.0 × 10^−3^) are key points that may be influencing metabolic or signaling pathways throughout the bovine mycoplasma system (Figure 6D).

By analyzing the subcellular localization of *M. bovis* proteins annotated and counted by cellular components (CCs) in the GO database, a total of 170 (152 shared, 18 unique) proteins of the two strains of *M. bovis* were localized to the membrane (NX114:64, PG45:68), cytoplasm (NX114:42, PG45:48) and ribosome (NX114:49, PG45:51) organelles (Figure 7). In addition, a total of 10 DEPs of *M. bovis* NX114 vs. PG45 were localized to the membrane (6 up-regulated, 4 down-regulated) and 4 DEPs were localized to the cytoplasm (1 up-regulated, 3 down-regulated) (Table 1).

### 3.4. Correlation Analysis of the Transcriptome and Proteome of M. Bovis

We further analyzed the correlation between the *M. bovis* NX114 and PG45 transcriptomes and proteomes, due to the inconsistency between the protein level and the transcription level. We extracted a total of 718 sequences from the identified transcripts and compared them with 905 identified protein sequences for sequence similarity. The results revealed that 631 (87.88%) CDS post-translational protein sequences of the transcripts aligned with a total of 870 (96.13%) proteins. A nine-quadrant association analysis revealed shared protein correlations (R = 0.44) with the same and opposite trend in shared gene/protein differential folds (Figure 8A). For genes that were significantly different in both proteomes and transcriptomes, we found a total of 49 protein/34 transcriptome CDS post-translational protein sequences, with the 15 DEGs and DEPs that were concurrently up-regulated in expression and the 21 DEGs and DEPs that were concurrently down-regulated being the most critical (Table 2, Figure 8B and Figure 9A).

We redrew the KEGG pathway annotation using the corresponding protein sequences of the transcriptome because the KEGG analysis of the transcriptome did not align with the chosen proteome reference. The shared KEGG analyses of the respective enrichment results revealed a total of 15 identical pathways (Figure 9B,C). The analysis of the proteome or transcriptome revealed eight significant pathways: metabolic pathways (mbv01100), glycolysis/gluconeogenesis (mbv00010), carbon metabolism (mbv01200), quorum sensing (mbv02024), biosynthesis of cofactors (mbv01240), microbial metabolism in diverse environments (mbv01120), biosynthesis of secondary metabolites (mbv01110), and ABC transporters (mbv02010) (Figure 9D). The energy metabolism pathways enriched the up-regulated DGPs TpiA and Pky among the six co-significant pathways (Figure 10)

### 3.5. qRT-PCR Verification

We selected some of the shared DEGs for qRT-PCR validation analysis, and calculated the expression of the corresponding genes in each group (2^−ΔΔCt^) using stably expressed GAPDH as an internal reference. The genes MBOVPG45_RS04010 (vspK), MBOVPG45_RS01760 (Lipoprotein), MBOVPG45_RS03700 (Lipoprotein), MBOVPG45_RS00040 (NADH-dependent flavin oxidoreductase), MBOVPG45_RS01765 (Lipoprotein), MBOVPG45_RS01755 (Lipoprotein), MBOVPG45_RS03705 (Glycerol ABC transporter, permease component), MBOVPG45_RS03865 (Ribosomal RNA small subunit methyltransferase I), MBOVPG45_RS03710 (Glycerol ABC transporter permease), MBOVPG45_RS01400 (HAD family hydrolase) and MBOVPG45_RS02835 (Lipoprotein) were down-regulated, and the genes MBOVPG45_RS01555 (Membrane lipoprotein P81), MBOVPG45_RS02985 (Lipoprotein_10 domain-containing protein) and MBOVPG45_RS00775 (Pyruvate kinase) were up-regulated (Figure 11). Although the degree of difference in gene expression varied between RNA-seq and qRT-PCR data, the DEGs, except MBOVPG45_RS01400 and MBOVPG45_RS02835, showed consistent expression trends, demonstrating the validity of the results of prokaryotic transcriptome and 4D-label-free quantitative proteomics analyses (Figure 11A).

In addition, the mRNA expression of TpiA and Pyk genes was not different between *M. bovis* NX114 vs. PG45 at the end of the delayed phase (12 h), but was significantly up-regulated at the end of the logarithmic phase (36 h) and at the persistence of the stabilized phase (48 h). The mRNA expression of TpiA and Pyk genes was not different during the decline phase (72 h) (Figure 11B).

## 4. Discussion

Transcriptomics and proteomics are two important tools for obtaining gene expression quantities. Proteomics provides data on the nature of the final gene products (proteins), complements comparative genomics and transcriptomics, and has a significant impact on microbiology in that it is able to generate proteomic profiles that can provide a detailed understanding of bacterial gene expression under specific conditions [17]. The biological functions of the transcription–protein dual-omics exhibit a clear backward and forward relationship, yet they are subject to influence. So, this study uses a combined transcription–protein dual-omics approach to look at the rules and nature of *M. bovis*’ life activities at different omics levels. It also shows how the two play a part in controlling each other and how the biology of different isolates fits together as a whole.

### 4.1. Important Components of Total Proteins of M. Bovis’ Clinical Strain

*M. bovis* has a size between that of bacteria and viruses, and does not have a nucleus and organelles such as mitochondria and Golgi apparatus, so its growth and metabolism need to obtain nutrients from the outside world [18]. We could clearly observe the cell membrane structure, the relatively low electron density of lipids comprising the intermediate layer, and the large number of ribosomes visible in both cytoplasms, as observed by transmission electron microscopy (Figure 1). The subcellular localization of this study showed that the membrane, cytoplasmic, and ribosomal proteins of the *M. bovis* NX114 clinical isolate are important components of total proteins. Among them, a total of 10 DEPs of *M. bovis* NX114 vs. PG45 were localized to the membrane, and 4 DEPs were localized to the cytoplasm (Table 1). Because there is no cell wall, *M. bovis* cell membranes can more fully fuse with host cells and exchange intracellular components [19]. This suggests that membrane proteins and cytoplasmic proteins play an important role in the interaction between *M. bovis* and its host. These key proteins serve a variety of biological functions, including acting as virulence factors, the invasion of the host, adhesion, signaling, nutrient uptake, immune system regulation, and toxic metabolite release [19].

### 4.2. Analysis of M. Bovis Adhesion Protein

Adhesins are key proteins in *M. bovis*-infected host cells that play important roles in pathogenic processes, including pathogen infection, cell invasion, colonization, immune escape, and virulence factors [20,21]. More than 10 species of *M. bovis* adhesins are known, which contain a variety of membrane surface proteins, some with immunogenic and conserved functions [22,23,24]. In addition, Plg, FN, HS and APLP2 proteins are now known to bind to *M. bovis* adhesins [20]. In this study, the *M. bovis* NX114 clinical isolate was found to contain six known adhesion proteins, three of which were differential proteins (Appendix A). In addition, a potential *M. bovis* adhesion protein (A0A059XYW1) was annotated in this study (Appendix A), and it remains to be verified whether it is involved in adhesion during *M. bovis* infection and promotes interactions with host–cell tight junctions.

### 4.3. Potential Virulence-Related Proteins Overrepresented in M. Bovis NX114 Whole Cell Proteins

To identify potential virulence-associated genes in *M. bovis* NX114, we used VFDB analysis to show that a total of 98 genes were associated with virulence in the two *M. bovis* strains, including 26 DEGs (11 up-regulated and 15 down-regulated) (Figure 3). Several studies have shown that P48, an important membrane protein associated with M. bovis virulence, can induce host–cell apoptosis through an endoplasmic reticulum stress-dependent signaling pathway [25]. This study found that P48 expression was downregulated. In addition, the nucleotide sequence of the *M. bovis* PG45 genome is known to have virulence-associated variable surface lipoproteins (Vsps), including VspE, VspI, VspO, VspJ, VspN, VspK, VspL, VspM, VspB, VspG, VspA, VspH, and VspF [26]. In this study, all 13 Vsps were jointly identified from two *M. bovis* strains, in which HYD67_03760, VspA, VspK, VspO, and VspJ were significantly down-regulated in DEPs and VspJ was lacking in *M. bovis* NX114, which indicated that the (Vsp) gene cluster differed less from *M. bovis* PG45 but differed more from *M. bovis* HB0801, which contained less than half of the Vsps [26]. The results of this study provide an important reference for further identification of potential virulence-related proteins in *M. bovis*.

### 4.4. Trend in Common DGPs and DEGs, and Pathways at All Levels of M. Bovis

The combined transcriptome–proteome analysis of *M. bovis* revealed a significant difference in 49 proteins out of 34 transcriptome CDS post-translational protein sequences, of which 15 were jointly up-regulated and 21 were jointly down-regulated. These proteins primarily included lipoproteins, chaperones, ABC transporters, structural domain proteins, ribosomal proteins, and key enzymes involved in glycolysis. The functional annotation and enrichment analyses of DEGs and DEPs revealed that DEPs were mainly enriched in metabolic pathways, glycolysis/gluconeogenesis, carbon metabolism, cofactor biosynthesis, microbial metabolism, cofactor biosynthesis in diverse environments, biosynthesis of secondary metabolites, and ABC transporters.

*M. bovis* has a limited biosynthetic capacity and requires nutrients from host cells to complete metabolism, as well as to evade host immune responses and damage cells [27]. The combined KEGG analysis revealed the potential for these proteins to exert their own virulence by influencing various metabolic pathways, thereby acquiring nutrient elements and producing *M. bovis* metabolites after a complex series of metabolisms in the host–cell. On the other hand, mycoplasma depends on the host for many nutrients and, thus, requires a robust transport system to transport nutrients [27]. Three types of transport systems have been identified in mycoplasma, including the ATP-binding cassette transporter (ABC), the phosphoenolpyruvate-dependent phosphotransferase system (PTS), and facile diffusion. Among them, the ABC transporter system is the main pathway for the exchange of substances between mycoplasma and the external environment [28,29]. In addition, it has been suggested that the substrate-bound ATP-dependent ABC transporter system involved in nucleoside uptake may be one of the surrogate transporters [28]. In the present study, it was shown that the ABC transporter system mediates micronutrient uptake in *M. bovis*, and it was found that the membrane protein P48 may be involved in the metabolism of *M. bovis* through the ABC transporter system, which can be utilized to carry out studies on *M. bovis* in this manner.

### 4.5. Key Enzymes Regulating Energy Metabolism and Growth in M. Bovis

Most mycoplasmas are unable to carry out a complete tricarboxylic acid cycle, so the glycolytic pathway serves as an important energy source for mycoplasmas, and it is of great significance to carry out research on enzymes related to the mycoplasma glycolytic pathway [30]. Triosephosphate isomerase (Tpi) and pyruvate kinase (Pyk) are two key enzymes in the glycolytic process carried out by mycoplasmas [30,31]. Tpi catalyzes the reversible conversion between dihydroxyacetone phosphate and D-type glyceraldehyde-3-phosphate, and Pyk is able to change phosphoenolpyruvate and ADP to pyruvate and ATP, making it one of the major rate-limiting enzymes in glycolysis [32]. In addition, pyruvate products produced by Pyk synthesis can regulate a variety of metabolic pathways. On the other hand, researchers have discovered that Tpi [31](encoded by the TpiA gene) and PK [33](encoded by the Pyk gene) of MG, along with numerous glycolytic enzymes in other mycoplasmas [34], are situated in the cell membrane and cytoplasm. These enzymes can play a role in adhesion to the host–cell, thereby exerting their harmful effects. In this study, TpiA was localized in the cytoplasm in *M. bovis* (Table 1), and TpiA (*p* < 0.001) and Pyk (*p* < 0.001) were significantly upregulated in *M. bovis* NX114 vs. PG45 (Table 2). Since most mycoplasmas lack genes related to the tricarboxylic acid cycle, the glycolytic pathway may be important for their energy acquisition. In this study, TpiA and Pyk were predominantly enriched in the energy metabolism (glycolysis/gluconeogenesis) and metabolic pathways (Figure 10), suggesting that the two enzymes regulate the metabolism of *M. bovis* and may be the source of the energy that provides for its survival and growth (reproduction, replication). The two strains of *M. bovis* in logarithmic phases showed different size morphologies, which presumably may be related to the activities of the two enzymes. Second, Tpi and PyK were enriched in the carbon metabolism pathway (Figure 10), and it was hypothesized that the two enzymes might be precursors of the carbon source of *M. bovis*. Because these two enzymes are less studied in *M. bovis*, they may provide research ideas for its pathogenic mechanism.

## 5. Conclusions

The study examined the biology of clinical isolates of *M. bovis* using transcriptomic and proteomic data. It identified 193 DEGs and 158 DEPs, and identified 49 proteins/34 transcriptomic CDS post-translational protein sequences. Membrane, cytoplasmic, and ribosome proteins were found to be significant components. *M. bovis* NX114 was associated with energy metabolism, secondary metabolite biosynthesis, and the ABC transport system. A new adhesion protein was annotated, and TpiA and Pyk genes were identified as key enzymes regulating *M. bovis* growth and maintenance. This study reveals the genes related to the regulation of *M. bovis* growth and provides research ideas for the pathogenic mechanism of *M. bovis*.

## Figures and Tables

**Figure 1 microorganisms-12-02012-f001:**
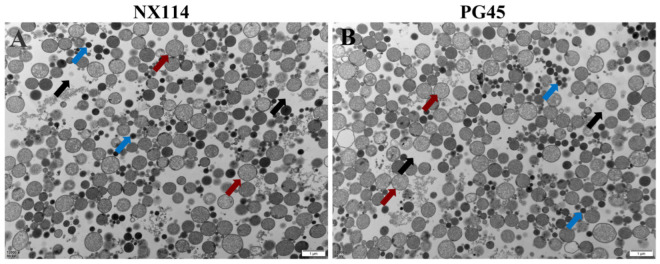
Organelles of *M. bovis* NX114 vs. PG45 transmission electron microscopy (12,000×). (**A**) *M. bovis* NX114. (**B**) *M. bovis* PG45. (black arrow: cytoplasm; blue arrow: soluble protein; red arrow: ribosome).

**Figure 2 microorganisms-12-02012-f002:**
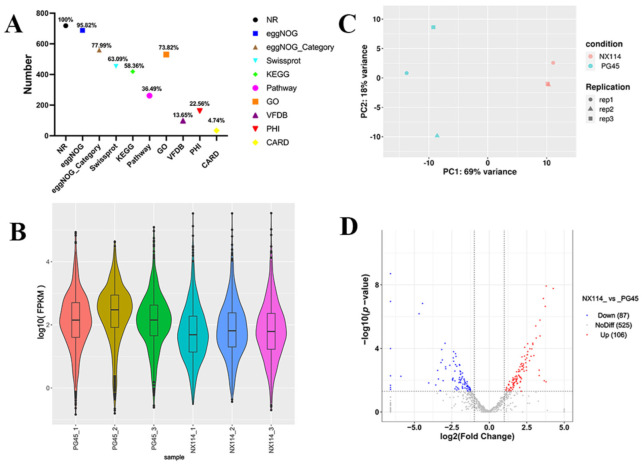
The general pattern of RNA-seq data and the identification of DEGs for *M. bovis* NX114 vs. PG45. (**A**) Genomic gene annotation information. (**B**) FPKM density distribution. (**C**) PCA analysis. (**D**) Volcano map of DEGs.

**Figure 3 microorganisms-12-02012-f003:**
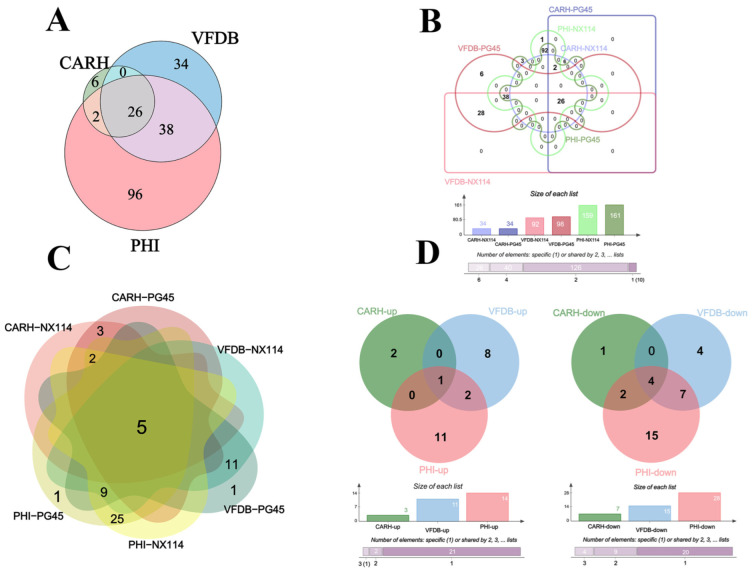
The annotation of the mRNAs of *M. bovis* NX114 vs. PG45. (**A**) Equal scale area Venn diagrams of the annotation information of the two strains of *M. bovis* together at CARH, VFDB and PHI. (**B**) Interactive Venn plots of the annotation information of the two strains of *M. bovis* together at CARH, VFDB and PHI. (**C**) The annotation information of DEGs of *M. bovis* NX114 vs. PG45. (**D**) Up- and down-regulated annotation information for DEGs of *M. bovis* NX114 vs. PG45.

**Figure 4 microorganisms-12-02012-f004:**
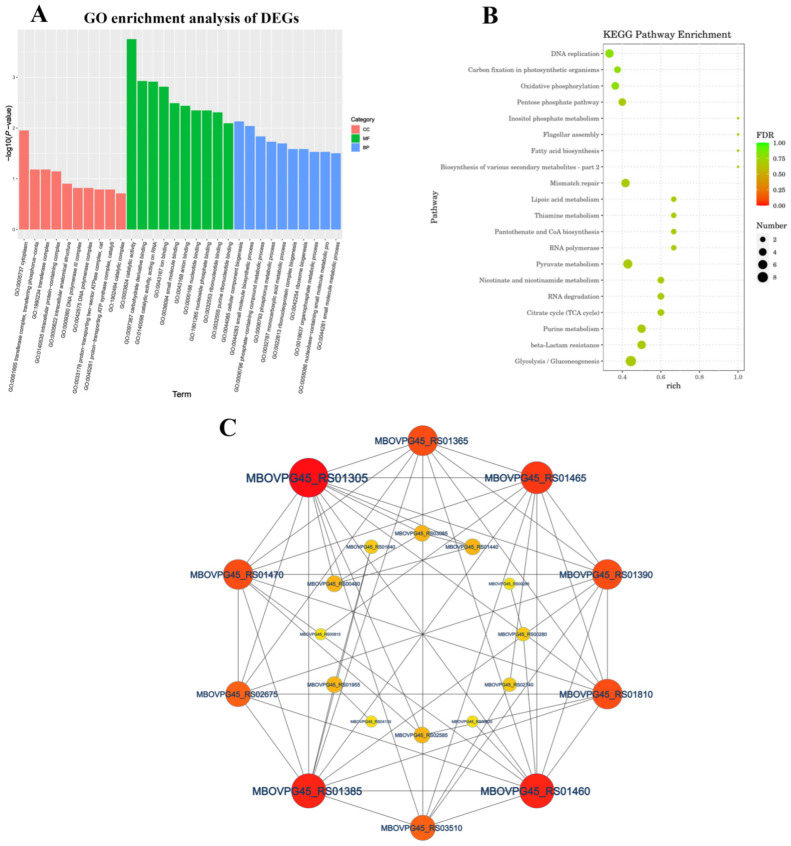
Functional categorization of DEGs of *M. bovis* NX114 vs. PG45. (**A**) GO enrichment analysis of DEGs. (**B**) KEGG enrichment analysis of DEGs. (**C**) PPi network analysis of DEGs.

**Figure 5 microorganisms-12-02012-f005:**
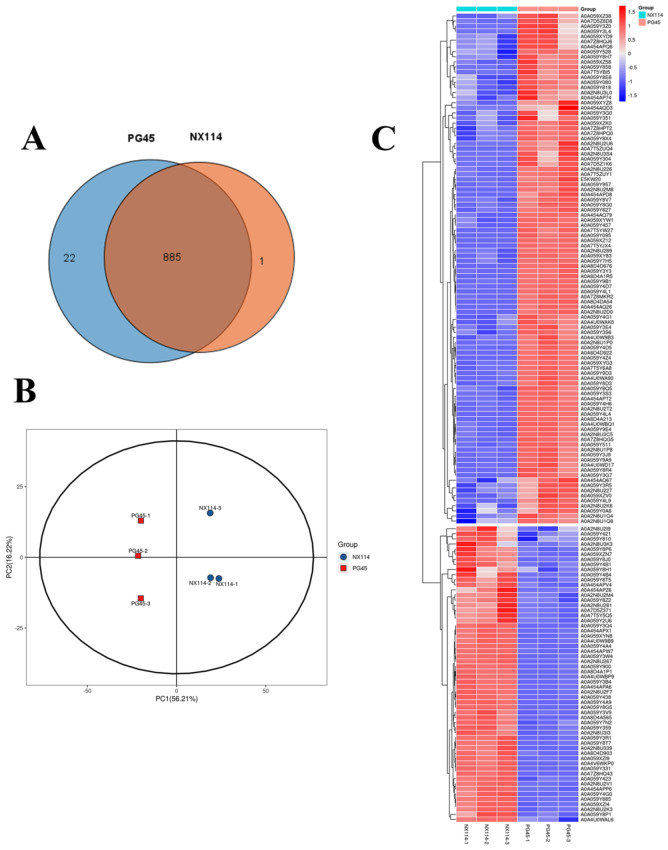
The proteomic analysis of *M. bovis* NX114 vs. PG45. (**A**) A Venn diagram of the proteome. (**B**) A PCA analysis of proteomic profiles. (**C**) A cluster analysis of DEPs of *M. bovis*.

**Figure 6 microorganisms-12-02012-f006:**
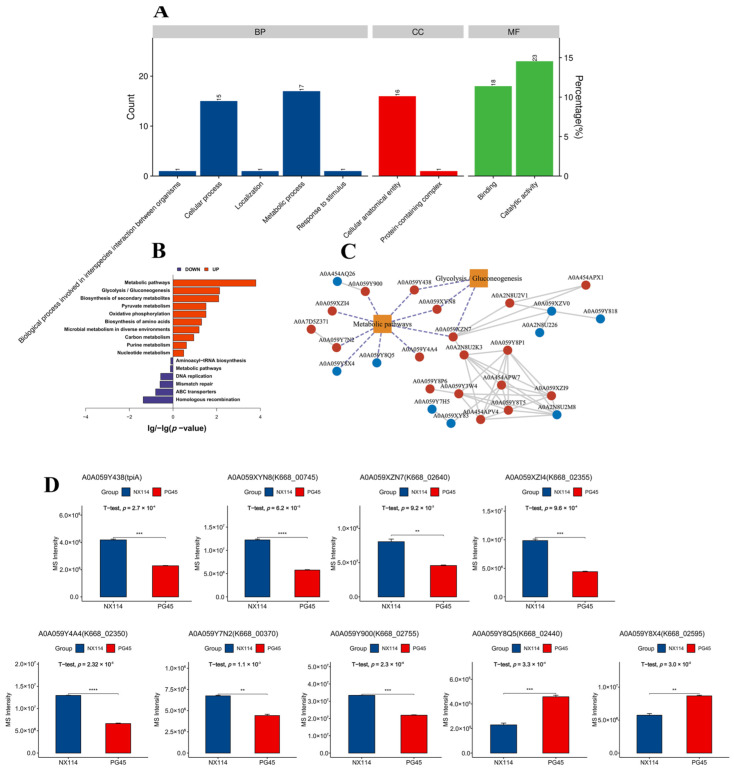
Functional categorization of DEPs of *M. bovis* NX114 vs. PG45. (**A**) GO annotation of DEPs. (**B**) KEGG enrichment analysis of DEPs. (**C**) Network analysis of significantly enriched pathways (*p* < 0.05) interacting with significantly different proteins, red vertices indicate up-regulated proteins, blue vertices indicate down-regulated proteins, and dotted lines represent key coding proteins. (**D**) Abundance analysis of key DEPs. ** *p* < 0.05, *** *p* < 0.01 and **** *p* < 0.001.

**Figure 7 microorganisms-12-02012-f007:**
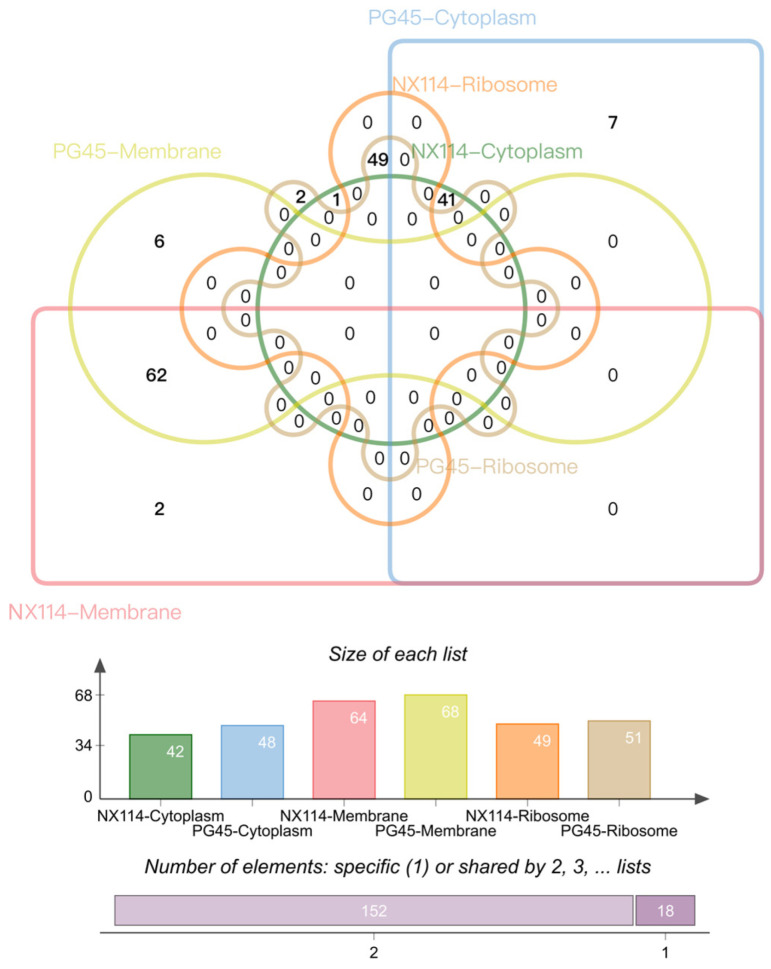
Results of CC subcellular localization of *M. bovis* NX114 vs. PG45 histones.

**Figure 8 microorganisms-12-02012-f008:**
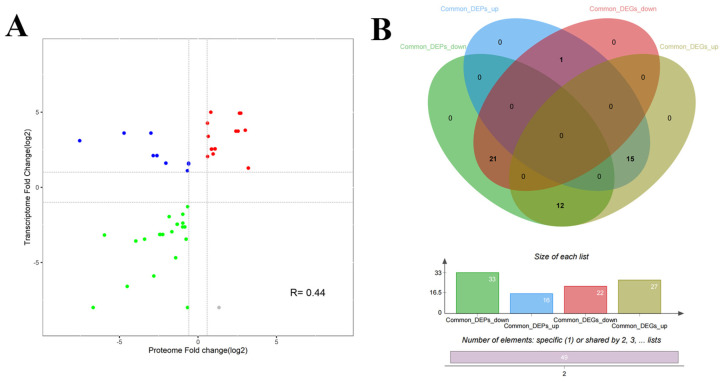
Statistical analysis of DGPs and DEGs of *M. bovis* (**A**) Common differential gene/protein differential fold correlation analysis. (**B**) Venn diagrams of DGPs and DEGs.

**Figure 9 microorganisms-12-02012-f009:**
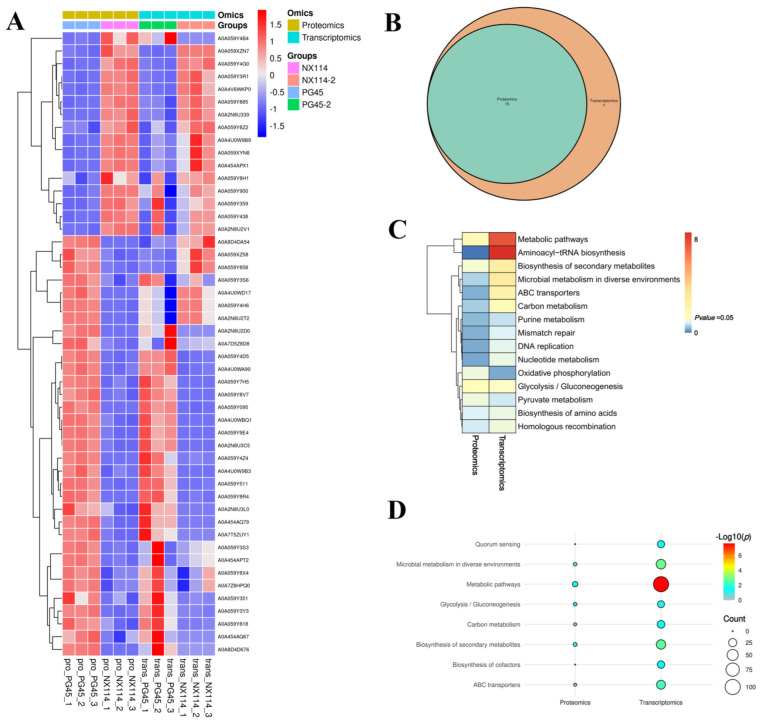
Joint analysis of *M. bovis* DGPs and DEGs. (**A**) Shared significant difference gene/protein clustering analysis. (**B**) KEGG comparison Venn diagram. (**C**) pvalue_heatmap. (**D**) Significant KEGG comparison bubble plot.

**Figure 10 microorganisms-12-02012-f010:**
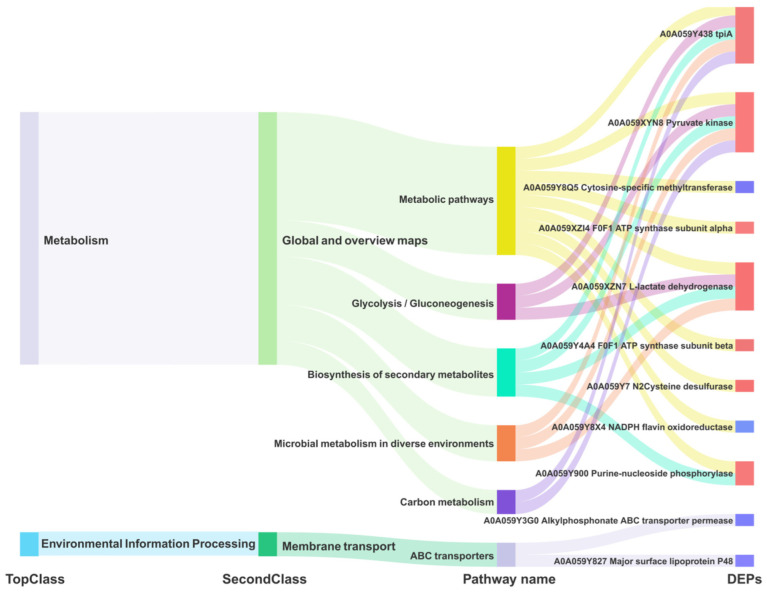
Dynamic Mulberry diagram of the common significant KEGG pathway of *M. bovis* DGPs and DEGs.

**Figure 11 microorganisms-12-02012-f011:**
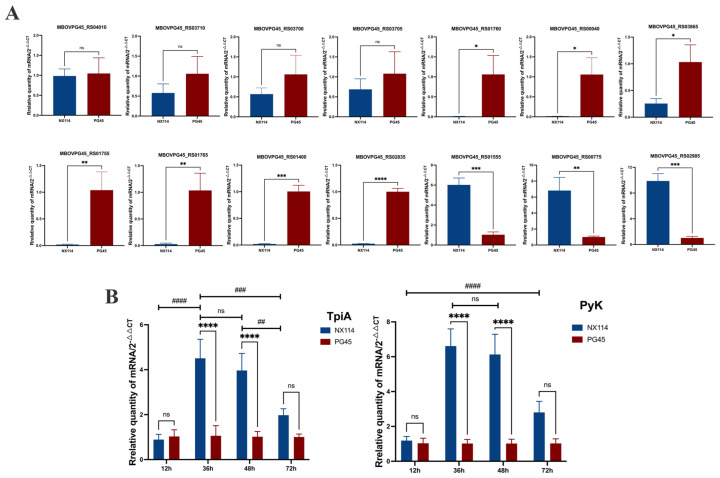
The validation of differentially expressed gene expression levels in *M. bovis* NX114 vs. PG45. (**A**) The validation of RNA-seq data for randomly selected genes by real-time PCR. * *p* < 0.05, ** *p* < 0.01, *** *p* < 0.001 and **** *p* < 0.0001 ns (no significance) *p* > 0.05. (**B**) qRT-PCR was used to detect the relative mRNA expression of TpiA and PyK genes at different growth stages. ## *p* < 0.01, ### *p* < 0.001 and #### *p* < 0.0001, ns (no significance) *p* > 0.05.

**Table 1 microorganisms-12-02012-t001:** The binding energy of compounds and core targets.

Query	Gene Symbol	Protein Name	FC	
A0A2N8U1Q4	MBOVJF4278_00156	Lipoprotein	0.647776028116661	Membrane
A0A2N8U289	BBB47_01840	Uncharacterized protein	0.426185200746393	
A0A2N8U2F7	BBB47_00485	DUF4064 domain-containing protein	4.48373919443482	
A0A2N8U2M4	MBOVJF4278_00496	Membrane-associated lipoprotein	2.11617261459845	
A0A454APD8	MBOVPG45_0232	Putative membrane protein (S41B peptidase family)	0.0737862069159731	
A0A454APP6	MBOVPG45_0130	Putative membrane protein	1.81544954347197	
A0A454APV4	MBOVPG45_0381	Putative membrane protein	1.84465678217211	
A0A454APW7	atpD-1	ATP synthase F1, beta subunit	1.95182048388406	
A0A4U0WAL6	MBOVa_6030	Uncharacterized protein	2.3262641252808	
A0A7D5Z8D8	H0I36_00720	Lipoprotein	0.627424403333434	
A0A2N8U2V1	tpiA	Triosephosphate isomerase	1.80897348893876	Cytoplasm
A0A454APQ6	polC	DNA polymerase III PolC-type	0.56429428286765	
A0A4U0WAK0	lysS	Lysine-RNA ligase	0.587245381151523	
A0A7Z8HQJ6	polC	DNA polymerase III PolC-type	0.56429428286765	

**Table 2 microorganisms-12-02012-t002:** DGPs and DEGs shared by *M. bovis* NX114 vs. PG45.

Query	Protein Name	Gene Symbol	Subject_ID	Q.Log2FC	S.Log2FC
A0A059XZ58	Lipoprotein	K668_01635	MBOVPG45_RS02835	−2.997308862	3.609877228
A0A059Y359	Lipoprotein	K668_00750	MBOVPG45_RS00780	0.964261054	2.221004141
A0A059Y3S6	Lipoprotein	K668_01310	MBOVPG45_RS02855	−0.597023412	1.601990172
A0A059Y4D5	Lipoprotein	K668_02500	MBOVPG45_RS01755	−1.317780028	−2.456487243
A0A059Y858	Lipoprotein	K668_01325	MBOVPG45_RS02835	−4.70915507	3.609877228
A0A059Y8R4	Lipoprotein	K668_02490	MBOVPG45_RS01765	−1.664446972	−2.954716109
A0A059Y8V7	Lipoprotein	K668_02495	MBOVPG45_RS01760	−1.420301578	−4.687636632
A0A059Y8Z2	Lipoprotein	K668_02705	MBOVPG45_RS01565	0.601695187	4.267861921
A0A2N8U2D0	Lipoprotein	MBOVJF4278_00420	MBOVPG45_RS02150	−2.813841488	−5.901035411
A0A2N8U3C5	Lipoprotein	MBOVJF4278_00737	MBOVPG45_RS03700	−2.405070273	−3.137107754
A0A4U0WA90	Lipoprotein	MBOVa_2490	MBOVPG45_RS01755	−1.33094428	−2.456487243
A0A4U0WBQ1	Lipoprotein	MBOVa_0550	MBOVPG45_RS03700	−2.24311657	−3.137107754
A0A7D5Z8D8	Lipoprotein	H0I36_00720	MBOVPG45_RS00720	−0.672486452	1.102694211
A0A059Y3R1	Lipoprotein_10 domain-containing protein	K668_01210	MBOVPG45_RS02985	2.642605673	4.934087976
A0A4V6WKP0	Lipoprotein_10 domain-containing protein	MBOVa_4800	MBOVPG45_RS02985	2.726388616	4.934087976
A0A059Y4H6	Membrane lipoprotein P81	K668_02710	MBOVPG45_RS01555	−2.614465995	2.111194499
A0A4U0WD17	P80, predicted lipoprotein	MBOVa_2120	MBOVPG45_RS01555	−2.852551282	2.111194499
A0A2N8U2T2	Putative lipoprotein MPN_284	MBOVJF4278_00566	MBOVPG45_RS01555	−2.614465995	2.111194499
A0A7T5ZUY1	Variable surface lipoprotein	HYD67_03760	MBOVPG45_RS04915	−3.959774718	−3.572406784
A0A454AQ79	Variable surface lipoprotein, VspK	vspK	MBOVPG45_RS04010	−3.402030599	−3.450026003
A0A059Y095	Glycerol ABC transporter permease	K668_03520	MBOVPG45_RS03710	−5.958377462	−3.175391715
A0A059Y3S3	ATPase	K668_02160	MBOVPG45_RS02110	−2.044928515	1.611376339
A0A059Y3Y3	ABC-2 type transporter ATP-binding protein	K668_02485	MBOVPG45_RS01770	−0.967247958	−2.381064151
A0A059Y4Z4	Glycerol ABC transporter, permease component	K668_03515	MBOVPG45_RS03705	−4.508651244	−6.594797599
A0A059Y9E4	Glycerol ABC transporter, glycerol binding protein	K668_03510	MBOVPG45_RS03700	−2.439239122	−3.137107754
A0A454APT2	ATPase	MBOVPG45_0422	MBOVPG45_RS02110	−2.044928515	1.611376339
A0A059Y511	Ribosomal RNA small subunit methyltransferase I	rsmI	MBOVPG45_RS03865	−1.83661242	−1.948913065
A0A059Y8H1	Ribosomal RNA small subunit methyltransferase E	K668_01990	MBOVPG45_RS02280	0.649478799	3.390515654
A0A059Y438	Triosephosphate isomerase	tpiA	MBOVPG45_RS01495	0.872625559	2.543542361
A0A2N8U2V1	Triosephosphate isomerase	tpiA	MBOVPG45_RS01495	0.855171265	2.543542361
A0A059XYN8	Pyruvate kinase	K668_00745	MBOVPG45_RS00775	1.087484925	2.559921425
A0A454APX1	Pyruvate kinase	pyk	MBOVPG45_RS00775	1.087484925	2.559921425
A0A4U0W9B9	Pyruvate kinase	pyk	MBOVPG45_RS00775	1.087484925	2.559921425
A0A059XZN7	L-lactate dehydrogenase	K668_02640	MBOVPG45_RS01630	0.816473286	4.999558053
A0A059Y351	Centromere protein F	K668_00230	MBOVPG45_RS00230	−0.967726122	−1.790262413
A0A059Y4B4	DUF31 domain-containing protein	K668_02400	MBOVPG45_RS01865	1.33879776	−8
A0A059Y4G0	Permease	K668_02635	MBOVPG45_RS01635	3.011818835	3.799031824
A0A059Y7H5	Trimethylamine dehydrogenase	K668_00045	MBOVPG45_RS00040	−0.838723456	−2.634300196
A0A059Y818	ResIII domain-containing protein	K668_00795	MBOVPG45_RS00815	−0.656365875	−1.285742333
A0A059Y885	Transposase	K668_01205	MBOVPG45_RS03450	2.553808449	3.73816659
A0A059Y8X4	NADPH flavin oxidoreductase	K668_02595	MBOVPG45_RS01660	−0.59617792	1.562983938
A0A059Y900	Purine-nucleoside phosphorylase	K668_02755	MBOVPG45_RS01510	0.609460858	2.051635587
A0A2N8U339	Transposase	H0I36_03435	MBOVPG45_RS03450	2.415751165	3.73816659
A0A2N8U3L0	Variant surface antigen A	MBOVJF4278_00819	MBOVPG45_RS04010	−0.755880306	−3.450026003
A0A454AQ67	Nucleotidyl transferase AbiEii/AbiGii toxin family protein	MBOVPG45_0815	MBOVPG45_RS04055	−0.664588662	−8
A0A4U0W9B3	NADH dependent flavin oxidoreductase	MBOVa_7200	MBOVPG45_RS00040	−0.979012158	−2.634300196
A0A7Z8HPQ0	Nitroreductase	BBB47_01090	MBOVPG45_RS01660	−0.59617792	1.562983938
A0A8D4D676	Abortive infection protein AbiGI	BC94_0718	MBOVPG45_RS04050	−6.67686922	−8
A0A8D4DA54	HAD family hydrolase	BC94_0552	MBOVPG45_RS01400	−7.535743554	3.103338051

## Data Availability

Data will be made available on request.

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
