# Peer review of "Integrating the Transcriptome and Proteome to Postulate That TpiA and Pyk Are Key Enzymes Regulating the Growth of Mycoplasma Bovis"

_microorganisms, 2024, doi:10.3390/microorganisms12102012_

Round 1

Reviewer 1 Report

Comments and Suggestions for Authors

Title: Integrating the transcriptome and proteome to speculation that TpiA and Pyk are key enzymes regulating the growth of Mycoplasma bovis

The paper compares the transcriptome and proteome of two Mycoplasma bovis strains, one virulent and the other the type strain. Also the authors postulate that  TpiA and Pyk genes may be the key enzymes that regulate the growth and maintenance of M. bovis and are involved in the pathogenic process as virulence factors. The paper is well executed and written and the results are going to be useful to scientists.

Specific comments:

I suggest to change the word speculation for postulate in the title.

L35: mycoplasmas instead of Mycoplasma

L52: do you mean Isobaric Tags for Relative and Absolute Quantification (iTRAQ™)?

L54: there is no reference Isobaric Tags for Relative and Absolute Quantification (iTRAQ™), please add at least one. There is a paper with mycoplasmas: Li S, Fang L, Liu W, Song T, Zhao F, Zhang R, Wang D, Xiao S. Quantitative Proteomic Analyses of a Pathogenic Strain and Its Highly Passaged Attenuated Strain of Mycoplasma hyopneumoniae. Biomed Res Int. 2019 Jul 1;2019:4165735. doi: 10.1155/2019/4165735.

L63: 4D-label-free quantitative proteomics instead of 4D-label-free quantitative non-labeled proteomics

L70 and L16: type strain instead of standard strain

L73: Which reference is the correct on 14 or Zhang R, Han X, Chen Y, Mustafa R, Qi J, Chen X, Hu C, Chen H, Guo A. Attenuated Mycoplasma bovis strains provide protection against virulent infection in calves. Vaccine. 2014 May 23;32(25):3107-14. doi: 10.1016/j.vaccine.2013.12.004?

L74-84 and Figure 1 and anything related to that figures: : I recommend removing it. Figures A and B does not offer anything new that already was published in: Yang, F.; Yang, M.; Si, D.; Sun, J.; Liu, F.; Qi, Y.; He, S.; Guo, Y. UHPLC/MS-Based Untargeted Metabolomics Reveals Metabolic Characteristics of Clinical Strain of Mycoplasma bovisMicroorganisms 202311, 2602. https://doi.org/10.3390/microorganisms11102602ç

And also, in figures 2 C and D, it is mentioned M. bovis organelles and I just see M. bovis cells

Figure 2 A: Number instead of Nunber

L248: Label-free or non-tagged. Label-free non-tagged is a redundancy

L369: be careful with the cursive letters, change it after M. bovis.

L443: mycoplasmas instead of Mycoplasmaf

 L472-473: Please rephrase the following sentence: The study revealed biological problems? in different M. bovis isolates and may provide research ideas

Author Response

Comments and Suggestions for Authors

General Comments: Title: Integrating the transcriptome and proteome to speculation that TpiA and Pyk are key enzymes regulating the growth of Mycoplasma bovis

The paper compares the transcriptome and proteome of two Mycoplasma bovis strains, one virulent and the other the type strain. Also the authors postulate that  TpiA and Pyk genes may be the key enzymes that regulate the growth and maintenance of M. bovis and are involved in the pathogenic process as virulence factors. The paper is well executed and written and the results are going to be useful to scientists.

Response: Thank you very much for taking the time to review this manuscript. We are very grateful to the reviewer for his/her favorable comments on our paper.

Secondly, we thank the referee comments, and responded to the reviewer′s comments point-to-point, as summarized below. All of the changes are marked in red font in the revised manuscript.

Other comments:

Specific comments 1:I suggest to change the word speculation for postulate in the title.

Response 1:We appreciate your helpful comments. According to the referees suggestion, we have revised the title of the article. The main changes are as follows:

“Integrating the transcriptome and proteome to postulate that TpiA and Pyk are key enzymes regulating the growth of Mycoplasma bovis

Comments 2: L35: mycoplasmas instead of Mycoplasma

Response 2:Thanks to the reviewer's carefulness, we have corrected this mistake(Lines L35).

Comments 3: L52: do you mean Isobaric Tags for Relative and Absolute Quantification (iTRAQ™)?

Response 3:Thanks to the reviewer's carefulness, we have corrected this mistake(Lines L53).

Comments 4: L54: there is no reference Isobaric Tags for Relative and Absolute Quantification (iTRAQ™), please add at least one. There is a paper with mycoplasmas: Li S, Fang L, Liu W, Song T, Zhao F, Zhang R, Wang D, Xiao S. Quantitative Proteomic Analyses of a Pathogenic Strain and Its Highly Passaged Attenuated Strain of Mycoplasma hyopneumoniae. Biomed Res Int. 2019 Jul 1;2019:4165735. doi: 10.1155/2019/4165735.

Response 4:Thanks for your suggestion, we have added this reference(Lines L54/L517-518).

Comments 5: L63: 4D-label-free quantitative proteomics instead of 4D-label-free quantitative non-labeled proteomics

Response 5:Thank you for reminding us. We have made the modification(Lines L63).

Comments 6:L70 and L16: type strain instead of standard strain

Response 6:We have made the correct changes(Lines L15-16 and L69).

Comments 7:L73: Which reference is the correct on 14 or Zhang R, Han X, Chen Y, Mustafa R, Qi J, Chen X, Hu C, Chen H, Guo A. Attenuated Mycoplasma bovis strains provide protection against virulent infection in calves. Vaccine. 2014 May 23;32(25):3107-14. doi: 10.1016/j.vaccine.2013.12.004?

Response 7:Thank you for your careful guidance. We found that these two articles by Professor Aizhen Guo both have relevant content, but the references you provided here are more authoritative in terms of publication time, so we replaced them(Lines L72/L524-525).

Comments 8:L74-84 and Figure 1 and anything related to that figures: : I recommend removing it. Figures A and B does not offer anything new that already was published in: Yang, F.; Yang, M.; Si, D.; Sun, J.; Liu, F.; Qi, Y.; He, S.; Guo, Y. UHPLC/MS-Based Untargeted Metabolomics Reveals Metabolic Characteristics of Clinical Strain of Mycoplasma bovis. Microorganisms 2023, 11, 2602. https://doi.org/10.3390/microorganisms11102602ç

And also, in figures 2 C and D, it is mentioned M. bovis organelles and I just see M. bovis cells

Response 8:We are grateful to the reviewer for his/her constructive comments.

We deleted the picture and related description of the negative staining part of mycoplasma in Figure 1, which was reported in our previous article (https://doi.org/10.3390/microorganisms11102602ç). Thank you again for your suggestion.

Second, we did not label the organelle part of M. bovis clearly, for which we annotated it (black arrow: cytoplasm; Blue arrow: soluble protein; Red arrow: ribosome)(Lines L171-176).

Comments 9:Figure 2 A: Number instead of Nunber

Response 9:We have made a change(Lines L86).

Comments 10:L248: Label-free or non-tagged. Label-free non-tagged is a redundancy

Response 10:Thanks to the reviewer's carefulness, we have corrected this mistake(Lines L238).

Comments 11:L369: be careful with the cursive letters, change it after M. bovis.

Response 11:Thanks to the reviewer's carefulness, we have corrected the sentence(Lines L365).

Comments 12:L443: mycoplasmas instead of Mycoplasmaf

Response 12:We have corrected the mistake(Lines L439).

Comments 13: L472-473: Please rephrase the following sentence: The study revealed biological problems? in different M. bovis isolates and may provide research ideas

Response 13: Thanks for your professional guidance, we have revised this sentence as follows:

“The study reveals the genes related to the regulation of M. bovis growth and provides research ideas for the pathogenic mechanism of M. bovis(Lines L468-470).

Reviewer 2 Report

Comments and Suggestions for Authors

The manuscript described a transcriptomic and proteomic data analyses of two Mycoplasma bovis strains. One isolated from cow with pneumonia in China and another is an international standard strain. They identified 193 and 158 differentially expressed genes and proteins, respectively. They performed an enrichment analysis for these genes and also performed a correlation analysis between transcription and protein expression profiles. However, it is not clear what the objective of the study is, as well as the experimental design adopted. Therefore, I have some comments for the authors to address:

Describe in more detail the strains used in the study, for example antibiotic resistance testing, virulence genes, etc. Explain in more detail why these two M. bovis strains were chosen?  What is the difference? For example, is the virulence different? Couldn't they have used a larger number of strains? The objective of the study is not clear.

The authors conclusion “A new adhesion protein was annotated, and TpiA and Pyk genes were identified as key enzymes regulating M. bovis growth and maintenance.” To reach this conclusion, it is necessary to confirm the regulation of these genes by performing experiments with mutated strains. The results of the present work may indicate the involvement of these genes and to improve this work, the authors could perform a growth curve and temporal expression analysis with qRT-PCR of TpiA and Pyk during the growth curve.

What do you mean by “histologies” and “histological levels” in lines 361 and 364? The context is unclear.

Fig. 3. B and C, please increase the letters and numbers.

Fig. 4 A and C, please increase the letters and numbers.

The figure captions could be better described to be self-explanatory.

Author Response

Comments and Suggestions for Authors

General Comments:The manuscript described a transcriptomic and proteomic data analyses of two Mycoplasma bovis strains. One isolated from cow with pneumonia in China and another is an international standard strain. They identified 193 and 158 differentially expressed genes and proteins, respectively. They performed an enrichment analysis for these genes and also performed a correlation analysis between transcription and protein expression profiles. However, it is not clear what the objective of the study is, as well as the experimental design adopted. Therefore, I have some comments for the authors to address: 

Describe in more detail the strains used in the study, for example antibiotic resistance testing, virulence genes, etc. Explain in more detail why these two M. bovis strains were chosen?  What is the difference? For example, is the virulence different? Couldn't they have used a larger number of strains? The objective of the study is not clear.

Response: Thank you very much for taking the time to review this manuscript. We think that the suggestions you have made are very professional and will be of obvious help to our research.

First, we explain why these two strains of Mycoplasma bovis (NX114 and PG45) were selected as follows:

We isolated a clinically pathogenic M. bovis strain (NX114) and sequenced its whole genome (GenBank accession no. CP135997). Using the new MLST method, we found that NX114 (ST52) and PG45 (ST12) belonged to different genotypes, and comparative genomic analysis with several epidemic strains of M. bovis showed that Mb-NX114 strain was significantly different from PG45 strain, and the degree of genomic collinearity was significantly lower than that of other strains. There are many gene inversions, deletions, and differences in distribution, which are not shown in this manuscript because they are in another manuscript that we wrote.

Meanwhile, in a previous report we used untargeted metabolomics to analyze the metabolic profiles of M. bovis NX114 versus PG45, and found that there were differences between the two strains in terms of growth phenotypes as well as metabolic profiles (Yang, F.; Yang, M.; Si, D.; Sun, J.; Liu, F.; Qi, Y.; He, S.; Guo, Y. UHPLC/MS-Based Untargeted Metabolomics Reveals Metabolic Characteristics of Clinical Strain of Mycoplasma bovis. Microorganisms 2023, 11, 2602. https://doi.org/10.3390/microorganisms11102602ç). Therefore, in this manuscript we would like to further expand the biological characterization of M. bovis by using combined transcriptome-proteome analysis in the hope of identifying key genes that can regulate the growth of M. bovis. In addition, your suggestion of antibiotic resistance testing and virulence gene testing is very important to us. In the following study, we will use multiple clinical isolates (different regions, different isolation sites and different genotypes) for analysis.

Secondly, we thank the referee comments, and responded to the reviewer′s comments point-to-point, as summarized below. All of the changes are marked in blue font in the revised manuscript.

Other comments:

Comments 1:The authors conclusion “A new adhesion protein was annotated, and TpiA and Pyk genes were identified as key enzymes regulating M. bovis growth and maintenance.” To reach this conclusion, it is necessary to confirm the regulation of these genes by performing experiments with mutated strains. The results of the present work may indicate the involvement of these genes and to improve this work, the authors could perform a growth curve and temporal expression analysis with qRT-PCR of TpiA and Pyk during the growth curve.

Response 1:Thank you very much for your professional guidance. We supplemented the mRNA expression analysis of TpiA and Pyk genes at different growth stages of M. bovis (mainly including 12h, 36h, 48h and 72h). The mRNA expression of TpiA and Pyk genes was not different between M. bovis NX114 vs PG45 at the end of the delayed phase (12h), but was significantly up-regulated at the end of the logarithmic phase (36h) and at the persistence of the sta-bilized phase (48h). The mRNA expression of TpiA and Pyk genes was not different during the decline phase (72h) (Figure 11B)(Lines L360). The results support the inference of the experiment(Lines L344-348). 

 Supplementary Figure:qRT-PCR was used to detect the relative mRNA expression of TpiA and PyK genes at different growth stages(Response to Reviewer).

Comments 2:What do you mean by “histologies” and “histological levels” in lines 361 and 364? The context is unclear.

Response 2:Thanks to the reviewer's carefulness, we have corrected this mistake. We replaced the “transcription-protein dual-omics”instead of“histologies”,the“dual-omicsinstead of “two omics” ,and the “omics levels” instead of “histological levels”(Lines L355-358).

Comments 3:

Fig. 3. B and C, please increase the letters and numbers.

Fig. 4 A and C, please increase the letters and numbers.

The figure captions could be better described to be self-explanatory.

Response 3:Thanks for your helpful comments. We have revised Fig. 3 (Lines L211)and Fig.4(Lines L235) . The figure captions were modified(Fig. 1:Lines L175-176ï¼›Fig. 6:Lines L281 and Fig. 11:Lines L362-364 ) . The title " GO enrichment analysis of DEGs" was added to Fig. 4 A(Lines L235).
